# O-GlcNAc Transferase Inhibitor Synergistically Enhances Doxorubicin-Induced Apoptosis in HepG2 Cells

**DOI:** 10.3390/cancers12113154

**Published:** 2020-10-27

**Authors:** Su Jin Lee, Oh-Shin Kwon

**Affiliations:** School of Life Science and Biotechnology, BK21 Plus KNU Creative BioResearch Group, College of Natural Science, Kyungpook National University, Daegu 41566, Korea; neojove79@knu.ac.kr

**Keywords:** O-GlcNAc transferase inhibitor, doxorubicin, apoptosis, P53, NF-κB, ER stress

## Abstract

**Simple Summary:**

We found that the combination treatment of doxorubicin (DOX) and O-GlcNAc transferase (OGT) inhibitor OSMI-1 has synergic therapeutic efficacy in the treatment of liver cancer. Our data show that DOX displayed cytotoxicity via the activation of p53 and the inflammatory NF-κB signaling pathway, while OSMI-1 evoked the ER stress response and inhibited NF-κB signaling. Therefore, DOX in combination with the OSMI-1 group showed a 20-fold reduction of tumor formation, whereas the DOX alone group reduced by 1.8-fold compared with control in a HepG2 cell xenograft model.

**Abstract:**

The combination of chemotherapy with chemosensitizing agents is a common approach to enhance anticancer activity while reducing the dose-dependent adverse side effects of cancer treatment. Herein, we investigated doxorubicin (DOX) and O-GlcNAc transferase (OGT) inhibitor OSMI-1 combination treatment, which significantly enhanced apoptosis in hepatocellular carcinoma cells (HepG2) as a result of synergistic drug action in disparate stress signaling pathways. Treatment with a low dose of DOX or a suboptimal dose of OSMI-1 alone did not induce apoptotic cell death in HepG2 cells. However, the combination of DOX with OSMI-1 in HepG2 cells synergistically increased apoptotic cell death through the activation of both the p53 and mitochondrial Bcl2 pathways compared to DOX alone. We also demonstrated that the combination of DOX and OSMI-1 stimulated cell death, dramatically reducing cell proliferation and tumor growth in vivo using a HepG2 xenograft mouse model. These findings indicate that OSMI-1 acts as a potential chemosensitizer by enhancing DOX-induced cell death. This study provides insight into a possible mechanism of chemotherapy resistance, identifies potential novel drug targets, and suggests that OGT inhibition could be utilized in clinical applications to treat hepatocellular carcinoma as well as other cancer types.

## 1. Introduction

Doxorubicin (DOX) is a chemotherapeutic agent widely used to treat solid tumors in liver and lung cancers [1]. However, DOX often displays reduced efficacy due to drug resistance, so new therapeutic treatments are urgently needed that can improve therapeutic effect and minimize side effects. DOX induces apoptosis in cancer cells by activating various caspases via DNA damage, ROS-induced damage, death receptor pathway activation, and the perturbation of mitochondrial function [2,3,4]. In this regard, the activation of the p53 signaling pathway is an essential signal in apoptotic cell death induced by DOX [5]. As a first-line tumor suppressor, p53 activates genes that promote apoptotic cell death as well as senescence and cell cycle control [6]. The ataxia-telangiectasia-mutated (ATM) kinase-dependent phosphorylation of p53 promotes the stability and transcriptional activation of p53 [7]. Under normal conditions, p53 expression is maintained at low levels by mouse double minute 2 homolog (MDM2) regulation, and p53 is rapidly stabilized upon DNA damage [8]. As a functional consequence, p53 initiates cell cycle arrest, apoptosis, and DNA repair through the induction of target genes such as *p21, Bax, and Puma* [9].

The hexosamine biosynthetic pathway (HBP) can be initiated in response to a variety of cellular stresses, and abnormal glycosylation via this pathway is directly related to altered cancer metabolism. O-GlcNAcylation is a post-translational modification that is regulated by the HBP. O-GlcNAcylation consists of the dynamic and reversible glycosylation of serine or threonine residues in various nuclear and cytoplasmic proteins [10]. The modification of O-GlcNAc onto the protein residue is catalyzed by the enzyme O-GlcNAc transferase (OGT), while its removal is catalyzed by the enzyme O-GlcNAcase (OGA) [11,12]. O-GlcNAcylation is increased in most malignant tumors, including liver cancer, where it positively correlates with tumor progression [13]. Although the detailed molecular mechanisms of regulation of these processes have not been clearly identified, there are many reports indicating that HBP-induced O-GlcNAcylation may be directly involved in DNA damage repair and cell survival as well as in resistance to DNA-targeted chemotherapy [14].

Interestingly, increased ER stress correlated with augmented protein O-GlcNAcylation. O-GlcNAc modification has been shown to reduce ER stress-induced cell death in cardiomyocytes [15]. ER stress is characterized by impaired protein folding, resulting in the accumulation of misfolded proteins [16]. To eliminate these misfolded proteins and restore ER homeostasis, the unfolding protein reaction, initiated by the saturation of BiP/GRP78, is triggered, resulting in the activation of protein kinase R (PKR)-like endoplasmic reticulum kinase (PERK), inositol requiring protein 1 (IRE1), and activating transcription factor 6 (ATF6) [17]. The expression of C/EBP homologous protein (CHOP) is upregulated via PERK/eIF2α signaling or the IRE-1/XBP-1 transcription factor pathway, which ultimately leads to apoptotic cell death [18,19]. CHOP promotes the inhibition of BCL-2, resulting in Bax-mediated permeation of the outer mitochondrial membrane [20]. This leads to cytochrome c release, and eventually to a caspase cascade, referred to as the intrinsic cell death pathway [21].

Numerous studies have shown that nuclear factor-kappa B (NF-κB) is often constitutively activated in human cancer cells [22,23]. NF-κB is a transcription factor that controls the expression of proteins involved in the regulation of cell survival and immune response [24]. NF-κB is activated by a variety of stimuli, including oxidative stress, ultraviolet light, and chemotherapeutic agents [25]. The canonical NF-κB dimer consists of the RelA/p65 and p50 subunits, which are bound by IκBα [26] in the cytosol. IκBα kinase (IKK) phosphorylates IκBα to induce its degradation, whereupon NF-κB is released and translocates to the nucleus and induces the expression of specific target genes [27]. RelA/p65 is post-translationally modified by phosphorylation or acetylation, which affects its transcriptional activity [28]. Interestingly, O-GlcNAc modification to p65 can also increase the nuclear translocation of NF-κB [29], but this mechanism is largely unclear.

A common approach to cancer therapy is to increase the anticancer activity of chemotherapeutics in combination with other drugs with different mechanisms of action while suppressing their undesired side effects. Here, we examined the therapeutic efficacy of the combination treatment of DOX and the OGT inhibitor OSMI-1 in liver cancer. This drug combination demonstrated synergistically increased anticancer activities by reducing cell proliferation and promoting apoptosis in vitro and in vivo. In particular, we demonstrated that DOX stimulates the p53 signaling pathway while the ER stress response is regulated by OSMI-1. We also determined that O-GlcNAc modulation of the NF-κB signaling pathway is directly linked to chemoresistance in HepG2 cells. These results suggest that the combination treatment of DOX and OSMI-1 has therapeutic potential in hepatocellular carcinoma (HCC) as well as other cancers.

## 2. Results

### 2.1. Combination Treatment with DOX and the OGT Inhibitor OSMI-1 Enhances Cell Death in HepG2 Cancer Cells

We initially investigated whether the extent of O-GlcNAcylation correlated with DOX-induced apoptosis in HCC cells. Although several reports have shown that the levels of *O*-GlcNAcylation are increased in many cancers, the association between O-GlcNAcylation levels and cell death in hepatomas is currently unclear. To address this, we examined the levels of this modification and its catalyzing enzyme OGT in cancer cell lines Hep3B, Huh7, and HepG2 and compared them to the noncarcinoma hepatocyte cell line AML12 (Figure 1A). In Huh7 and HepG2 cells, p53 expression, as well as *O*-GlcNAc modification and OGT levels, appeared to be higher than in Hep3B and AML12 cells. We also determined whether OSMI-1, a pharmacological inhibitor of OGT, could stimulate cell death. MTT assays were performed to determine the viability of liver cells upon low-dose DOX treatment (0.4 uM) alone or in combination with OSMI-1. As shown in Figure 1B, the difference in the viability between DOX-treated and OSMI-1 combination-treated cells was more significant in HepG2 cells than in other cell lines. The correlation between viability and *O*-GlcNAcylation levels between cells requires further research to be confirmed, but in this study, we focused on the cell death resulting from the combination treatment in HepG2 cancer cells.

We next measured the viability of HepG2 cells in response to a range of DOX and OSMI-1 concentrations. As shown in Figure 1C (first panel), the cell viability in response to DOX decreased in a concentration-dependent manner, but the effect was insignificant at low concentrations in the 0.1–1 μM range, not reaching below 80% survival. However, when OSMI-1 (20 μM) was also used, the concentration dependence of DOX dramatically increased, and the proliferative activity of HepG2 was significantly inhibited by about a further 30%. On the other hand, OSMI-1 treatment alone had little effect on cell viability. OGT siRNA was used to confirm that the concentration-dependent cell viability of DOX was linked to the level of OGT expression (Appendix A). As in the results following the DOX/OSMI-1 combination treatment, the combination treatment of OGT knockdown with DOX dramatically reduced cell viability. Similar results were also found in the concentration dependence of OSMI-1 (Figure 1C, second panel). The concentration dependence of OSMI-1 treatment alone was much lower than that of DOX and had little effect on survival in comparison to the combination treatment with DOX. Treatment with 0.4 μM of DOX alone as a control slightly reduced the level of cell viability. These data suggest that the combination of OSMI-1 and DOX synergistically inhibits proliferation and promotes cell death in HepG2 cells. To confirm these results, a cell colony formation assay was also used to determine the proliferation of HepG2 cells after treatment with DOX alone or in combination with OSMI-1 (Figure 1D). After cultivation for 8–10 days, cell colonies were visualized by crystal violet staining. Similar to the results of the MTT assays, the formation of colonies was significantly reduced after the combination treatment, suggesting that combination therapy was much more effective than monotherapy.

### 2.2. OSMI-1 Enhances DOX-Induced Apoptotic Cell Death via p53 Activation

DOX has been shown to induce p53 overexpression in liver cancer cells in vivo [30]. To evaluate the apoptotic effect of DOX in HepG2 cells, various concentrations of DOX alone or in combination with OSMI-1 were used for 15 h (Figure 2A). Levels of p53, along with p21, were upregulated in a DOX-dose-dependent manner (upper panel). However, no effect on p53 expression was detected with OSMI-1 treatment alone (lower panel). We further evaluated whether the effect of the OSMI-1-induced sensitization of HepG2 cells to DOX-induced apoptosis was dose-dependent. As shown in Figure 2B, treatment with 0.4 μM of DOX alone did not obviously induce apoptosis, whereas cotreatment with OSMI-1 dramatically increased cleaved caspase-3 signal in a dose-dependent manner. In addition, these results suggest that OSMI-1 further enhanced p53 expression over what was induced by DOX alone. Interestingly, the increases in p53 expression and proapoptotic PARP signal were confirmed in the combination treatment of HepG2 cancer cells, but no significant changes were observed in AML12 normal cells (Figure 2C). Taken together, these results indicate that OSMI-1 can enhance DOX-induced cell apoptosis in HepG2 cells but not in AML12 normal cells. The low doses of DOX (0.4 μM) and OSMI-1 (20 μM) were subsequently used in all further cell culture experiments, considering the minimal toxicity and therapeutic effect observed.

To confirm whether p53 was regulated by the DOX/OSMI-1 combination treatment resulting in apoptosis, we used the ATM inhibitor ku55933 and p53 siRNA (Figure 2D). It is well known that when cellular DNA is damaged by DOX treatment, ATM is phosphorylated and activated, leading to eventual increased p53 expression. In this study, the levels of p-ATM and PARP were increased with DOX treatment alone and increased significantly further upon the combination treatment with OSMI-1. However, the effects on the downstream-effector-cleaved PARP were decreased by treatment with the ATM inhibitor ku55933 (left panel). Similarly, treatment with p53-specific siRNA significantly reduced the signal of the enhanced apoptotic cell death marker (right panel). These results indicate that the apoptotic effect of the DOX/OSMI-1 combination treatment is directly related to p53 signaling. It is also known that ATM is modified by O-GlcNAc, which negatively regulates its activation through competition for the same phosphorylation residue. As shown in Figure 2E, treatment with DOX alone clearly induced the phosphorylation of ATM, while cotreatment with OSMI-1 dramatically enhanced the phosphorylation level of ATM compared to nontreated HepG2 cells. We also analyzed the expression levels of enzymes that govern O-GlcNAcylation, such as OGT and OGA. OGA levels were almost constant regardless of treatment with DOX or OSMI-1, but OGT expression was upregulated by DOX and more significantly upregulated by the combination treatment of DOX/OSMI-1. These changes in the expression of OGT appear to contribute to maintaining cellular homeostasis.

### 2.3. OSMI-1 Induces the ER Stress Response

We first examined the basal expression levels of PERK and IRE1α proteins in the absence or presence of OSMI-1 in liver cell lines (Figure 3A). PERK and IRE1α were upregulated in OSMI-1-treated HepG2 cells, whereas no significant changes were observed in AML12 cells. These data indicate that OSMI-1 induces the activation of PERK and IRE1α-XBP1 signaling, especially in hepatocellular carcinoma cell lines. We then investigated whether the inhibition of O-GlcNAcylation modulates cell death signaling via the ER stress response in HepG2 cells. As shown in Figure 3B, changes in O-GlcNAcylation levels were not observed after treatment with a low dose of DOX but were significantly decreased in a concentration-dependent manner following OSMI-1 treatment. Contrary to the change in O-GlcNAcylation levels, the expression of PERK and IRE1α-XBP was slightly increased by DOX treatment but was significantly upregulated in a concentration-dependent manner by OSMI-1. The downstream target Bcl2 was downregulated by both DOX and OSMI-1 treatment. We further evaluated the mechanism by which OSMI-1 sensitized cells to DOX-induced apoptosis. As shown in Figure 3C, treatment with 0.4 μM of DOX alone decreased Bcl2 levels slightly but failed to increase Bax, resulting in insufficient apoptotic cell death. However, the combination treatment of DOX and OSMI-1 dramatically increased cleaved caspase-3 and PARP in HepG2 cells. O-GlcNAcylation levels showed no significant changes in DOX-treated cells but were significantly reduced upon treatment with DOX and OSMI-1 together. These results indicate that OSMI-1 can stimulate DOX-induced cell apoptosis. CHOP is also known to promote mitochondria-mediated apoptosis through the downregulation of the prosurvival protein Bcl2 [31]. In this study, we showed that CHOP mRNA levels gradually increased in a concentration-dependent manner according to OSMI-1 (Figure 2D). Contrary to CHOP, levels of Bcl2 were significantly decreased by OSMI-1. Taken together, these results suggest that in the presence of DOX, OSMI-1 synergistically enhances the activation of the ER stress response, leading to apoptosis via the Bcl2/Bax pathway.

### 2.4. NF-κB Signaling Is Modulated by OSMI-1 in DOX-Treated HepG2 Cells

NF-κB is important in regulating genes in response to oxidative stress. These genes contribute to the inflammatory response [32,33]. We examined the activation of the NF-κB p65 subunit with respect to the treatment duration of DOX (0.4 μM) in HepG2 cells (Figure 4A). As shown in the left panel, the levels of phospho-p65 increased and peaked after 60 min and decreased thereafter. In addition, p65 levels were compared after treatment with DOX alone and in combination with OSMI-1 (20 μM) for 60 min (right panel). The levels of phosphorylated p65 gradually increased with time in DOX-treated cells, but phosphorylation was completely blocked in the cells treated with OSMI-1. These results indicate that DOX-induced damage initially resulted in inflammation or cell survival but was inhibited in the presence of OSMI-1. As a result, DOX-induced p65 activity compensated for the negative regulation by OSMI-1, suggesting that p65 activity is directly or indirectly related to O-GlcNAc modification. The activation of NF-κB and p53 signaling were also dependent on DOX concentration (Figure 4B). NF-κB signaling upstream effectors, such as p65, IKK, and IκB, were gradually activated following a low dose of DOX (maximum at 0.4 μM), but at higher concentrations, their activities were gradually decreased. Unlike these, p53 expression level was consistently upregulated with DOX treatment up to 4 μM. Taken together, NF-κB signaling is activated along with p53 signaling following treatment with low concentrations of DOX. However, as the concentration-dependent DOX-induced damage was increased, the influence of NF-κB signaling gradually decreased, resulting in a dramatic increase in p53 signaling.

To further confirm the involvement of O-GlcNAcylation in DOX-induced NF-κB signaling, HepG2 cells were transfected with either control or OGT siRNA and treated with DOX (0.1 and 0.4 μM). As shown in Figure 4C (upper panel), the activation of p65 and IKK in OGT knockdown cells was completely inhibited, while a concentration-dependent increase occurred in control siRNA-transfected cells. It has been shown that IKK can also be O-GlcNAcylated by OGT [34]. To determine if the activation of IKK by DOX treatment is related to O-GlcNAcylation, the levels of O-GlcNAcylated IKK were analyzed by immunoblotting after immunoprecipitation (lower panel). While the level of GlcNAc modification of IKK proportionally increased with the DOX concentration, it was not detected at all in the cells transfected with OGT siRNA. These results indicate that O-GlcNAc modification of the IKK protein increased in a DOX-concentration-dependent manner.

Meanwhile, there are several reports that p65 accumulates in the nucleus upon stimulation of cells with DOX. Thus, we investigated whether the O-GlcNAc modification of p65 affected its nuclear localization (Figure 4D). DOX treatment rapidly stimulated the nuclear translocation of p65, but treatment with OSMI-1 significantly prevented p65 nuclear translocation. On the contrary, as expected, p53 expression levels were upregulated by DOX treatment alone and increased further with the OSMI-1 combination treatment. Similar to these results, we also used immunofluorescence to confirm that the levels of nuclear p53 were significantly increased upon cotreatment with OSMI-1 rather than with DOX alone (Appendix A). Next, to confirm the O-GlcNAcylation of p65, immunoprecipitation with the p65 antibody was performed followed by immunoblotting with the O-GlcNAc antibody. The level of O-GlcNAc modification of p65 was increased after DOX treatment but decreased after OSMI-1 treatment. These results indicate that O-GlcNAc modification of p65 is associated with its nuclear translocation. We further evaluated inflammation and apoptotic cell death in HepG2 cells treated with DOX or transfected with OGT siRNA (Figure 4E). IL-1 and iNOS were upregulated by DOX treatment in control siRNA-transfected cells, but this increase upon DOX treatment was not observed in OGT knockdown cells. On the other hand, Bcl2 and proapoptotic signals such as cleaved PARP and caspase-3 were slightly reduced or hardly detectable after DOX treatment in the control cells. However, in the cells transfected with OGT siRNA, the decrease in Bcl2 was remarkable, and cleaved PARP and caspase-3 were clearly identified. Taken together, these results suggest that the DOX-induced cell damage is partially alleviated by NF-κB activity, which is consistent with drug resistance activity. Thus, the combination treatment of OSMI-1 with DOX may significantly reduce drug resistance, resulting in enhanced cell damage, and ultimately the synergic increase of apoptotic cell death.

### 2.5. Combination Treatment with OSMI-1 and DOX Synergistically Enhanced Anticancer Activity in HepG2 Xenograft Mice

Based on the observed results in vitro, we next addressed whether the double inhibition of NF-κB and OGT would increase the efficacy of DOX in vivo. To evaluate whether the combination treatment with low-dose DOX (0.1 mg/kg) and OSMI-1 (1 mg/kg) could be utilized as a potential therapeutic treatment, we used a subcutaneous HepG2 xenograft mouse model. HepG2 cells were injected subcutaneously into the left flank of nude mice and tumors were palpable at the injection sites after seven days. The mice were divided into four treatment groups. After administration of DOX or OSMI-1 alone or in combination for 32 days, the tumor volume was measured at different time points (Figure 5A); representative images showing the size of the extracted tumor are shown in the right panel. DOX in combination with the OSMI-1 group showed a 4.5-fold reduction of tumor formation, whereas mice treated with OSMI-1 or DOX alone did not differ from the control group. Western blot analysis indicated the levels of proteins associated with apoptotic cell death in xenografts (Figure 5B). Compared with vehicle-treated xenograft mice, the expression of p53 and phospho-p65 was increased upon DOX treatment alone but slightly decreased or did not change at all upon OSMI-1 treatment alone. However, in the combination treatment, p53 level was increased much more than with DOX alone, while active p65 levels returned to a level similar to the control group, demonstrating the synergistic activation of the cell death pathway. These results are consistent with the previous in vitro experiments, indicating that p65 signaling activity decreases due to the compensatory actions of DOX and OSMI-1, and as a result, apoptotic cell death was synergically increased via the p53 signaling pathway.

To further validate these results, we also explored the effectiveness of higher doses of chemotherapy. To evaluate whether the combination treatment with high-dose DOX (1 mg/kg) and OSMI-1 (5 mg/kg) could be utilized as a potential therapeutic treatment and the tumor volume was measured at different time points (Figure 5C); representative images showing the size of the extracted tumor are shown in the right panel. DOX in combination with the OSMI-1 group showed a 20-fold reduction of tumor formation, whereas the DOX alone group was reduced by 1.8-fold compared with the control in the HepG2 cell xenograft model. Moreover, no other signs of acute or delayed toxicity were observed in the mice during treatment. Western blot analysis showed an increase in the levels of proteins associated with apoptotic cell death in xenografts (Figure 5D). As compared with vehicle-treated mice, the expression of p53 and Bax was increased upon DOX or OSMI-1 treatment alone but was much more increased following the combination treatment. p21 and cleaved caspase-3 were upregulated in the DOX alone and combination treatment groups but levels were basal following OSMI-1 treatment alone as in the control. In particular, there was a dramatic increase in cleaved caspase-3, indicating apoptotic cascades were occurring only in the combination treatment group. In addition, as expected, the expression levels of IRE-1 and PERK were significantly increased with OSMI-1 treatment. To characterize the phenotype of the xenografts after each treatment, immunohistochemistry (IHC) analysis was performed (Figure 5E). The cell proliferation marker Ki-67 showed that the proliferation of tumor cells slightly decreased upon DOX or OSMI-1 treatment alone, but dramatically decreased with the combination treatment. As predicted from the Western blot results, in the combination treatment with DOX and OSMI-1, the expression levels of cleaved caspase-3, p53, and PERK showed the highest increases compared to other treatments, while on the contrary, the levels of nuclear p65 were the lowest in this group. Upon DOX alone treatment, the level of nuclear p65 expression was highest, while the levels of PERK were not changed. In addition, in the treatment of OSMI-1 alone, changes in p65 and PERK were easily observed, but no apoptosis-related signal was detected at all.

Taken together, we propose that the combination treatment of DOX and OSMI-1 synergistically induce apoptotic cell death via ROS production and ER stress induction (Figure 5F). The main downstream effectors of each drug were the activation of p53 and PERK/IRE-1, whereas NF-κB signaling was either regulated by ER stress or simultaneously inhibited p53 signaling.

## 3. Discussion

The combination treatment of anticancer drugs and supplements with different mechanisms of action is a commonly used strategy to increase therapeutic efficacy while avoiding side effects [35,36]. The main goal of this study was to determine if OGT inhibition was able to enhance the therapeutic efficacy of DOX in the treatment of liver cancer. We examined whether the combination treatment of DOX with OGT inhibitor OSMI-1 enhanced anticancer activities without aggravating side effects in HepG2 cells. Indeed, the cotreatment of DOX with OSMI-1, both of which interfere with cell survival, showed synergistically enhanced anticancer activity in vitro and in vivo. In this study, we primarily focused on the role of the OGT inhibitor in the combination treatment and the evaluation of the key signaling pathways involved. The inhibition of O-GlcNAcylation by OSMI-1 increased ER stress and blocked NF-κB signaling, resulting in a significant decrease in chemoresistance. Thus, OSMI-1 appears to act as a potential chemosensitizer, enhancing DOX-induced cell death.

DOX induces apoptotic cell death in cancer cells through the production of ROS, DNA damage, cell cycle arrest, and p53 activation [37,38]. In the present study, a low dose (0.4 μM) of DOX was used for the treatment of HCC, resulting in more than 80% cell viability. However, treatment with a combination of DOX and OSMI-1 enhanced apoptotic signaling via the p53 pathway and thereby improved the therapeutic efficacy of DOX by approximately 50%. The inhibition of O-GlcNAcylation in cancer cells has been suggested to decrease chemoresistance [39]. ATM activation has been also reported to be negatively regulated by O-GlcNAc modification, resulting in suppression of p53 target genes [40]. In this study, we confirmed that OSMI-1 treatment increased ATM phosphorylation and subsequently upregulated p53 to induce apoptosis in DOX-treated HepG2 cells. It has been previously shown that cytoplasmic p53 participates in the intrinsic death pathway by directly interacting with Bcl2-family proteins to induce mitochondrial outer membrane permeation [41].

O-GlcNAcylation is increased in various cellular stresses, including chemotherapy treatment and ER stress, directly influencing cell survival in cancer cells. It has been reported that DOX activates the HBP through the AKT/XBP axis, which can promote O-GlcNAcylation [42]. Moreover, a recent study has suggested that reducing O-GlcNAcylation in cancer cells leads to CHOP induction through the activation of the ER stress response, resulting in apoptosis [43]. In this study, we demonstrated that OGT inhibitor OSMI-1 enhances ER stress, leading to increased IRE1-XBP and PERK-CHOP signaling in HepG2 cells (Figure 3). CHOP mRNA, a prominent marker of ER stress-initiated apoptotic signaling, was upregulated in a dose-dependent manner by OSMI-1. On the other hand, the antiapoptotic gene Bcl2 was reversely inhibited by the increased CHOP, and as a result, Bax expression was increased. Interestingly, even at a relatively low concentration, treatment with a combination of DOX and OSMI-1 synergistically promoted ROS production and ER stress, resulting in a dramatic increase in caspase-3 activation and apoptosis. However, mitochondrial damage induced by OSMI-1 treatment alone rarely led to cell death.

Recently, emerging data have revealed that constitutive activation of NF-κB is one of the key mechanisms of chemoresistance [44]. Therefore, the inhibition of the NF-κB pathway may enhance the efficacy of cancer therapy. Nevertheless, whether NF-κB is involved in doxorubicin resistance in HCC remains poorly understood. OGT inhibition has been reported to inhibit proliferation and tumor drug resistance by directly affecting the NF-κB pathway in HCC [45,46]. In this study, we found that DOX-induced NF-κB activation was inhibited by OSMI-1, and subsequently increased cell death (Figure 4). In DOX-treated HepG2 cells, p65 and IKKβ activities were found to be dependent on DOX concentration. However, in the presence of OSMI-1, these activities were significantly reduced to a basal level. These results were confirmed by IP and siRNA analyses, indicating that the O-GlcNAcylation of p65 and IKK positively correlates with the level of activity. Here, it is important to note that NF-κB was activated by DOX-induced ROS, while it was negatively regulated by OSMI-1. Therefore, blockade of the induced NF-κB in this combination treatment may counteract the survival of cancer cells and therefore promote apoptotic cell death. The regulation of GlcNAcylation of NF-κB in HCC cells may be directly related to DOX resistance, although further study of this mechanism is needed.

Taken together, we showed here that the combination treatment of OSMI-1 and DOX synergistically increased cancer-specific cytotoxic activity through at least three parallel mechanisms. First, the increase of ROS by DOX led to increased p53 signaling; second, the ER stress response was induced by the OGT inhibitor OSMI-1; and finally, NF-κB signaling related to cell survival and inflammation was decreased. As a result, Bcl2 levels were decreased, resulting in the cleavage of caspase-3 and ultimately apoptotic cell death. On the other hand, treatment with DOX or OSMI-1 alone initiated the respective arms of this signaling network but did not lead to apoptotic cell death. These results show that crosstalk between inflammation and ER stress contributes to the decision between cell survival and death. IRE-1 is known to activate NF-κB by causing IkB degradation, whereas PERK negatively regulates the proinflammatory process through CHOP. Therefore, we suggested that NF-κB regulation is induced in response to ER stress, an important determinant, which influences sensitivity to p53. The combination treatment in the HepG2 xenograft mouse model dramatically lowered the volume and growth of liver tumors in vivo. To our knowledge, this is the first report in which OSMI-1 acts as a chemosensitizer when cotreated with DOX in a xenograft mouse model, dramatically improving the therapeutic efficacy of DOX in liver cancer. Another notable point is that this combination treatment displayed higher synergy when the chemotherapeutic agent was used at a low concentration. Our results suggest that at high DOX concentrations, the ER stress pathway is blocked, which neutralizes the effect of OSMI-1, leading to unilateral DOX-concentration-dependent cell death. Therefore, it is necessary to find an appropriate threshold to maximize therapeutic efficacy so as to lower resistance to chemical sensitizers while maintaining low cytotoxicity.

## 4. Materials and Methods

### 4.1. Preparation of Cell Lysates and Western Blot Analysis

AML12, Hep3B, Huh7, and HepG2 cells were extracted using NP-40 lysis buffer containing 50 mM HEPES (pH 7.4), 150 mM NaCl, 1 mM EDTA, 1% NP-40, and protease inhibitor cocktail tablets (Roche, Mannheim, Germany). Proteins were separated by electrophoresis on 8–15% SDS-PAGE gels and transferred to a nitrocellulose membrane (GE Healthcare Life science). After blocking with 5% skimmed milk at room temperature for 1 h, membranes were incubated with primary antibodies at 4 °C overnight. The extracts were subjected to Western blot using antibodies directed against O-GlcNAc (Sigma-Aldrich, St Louis, MO, USA), OGA, IL-1β (Abcam, Cambridge, UK), PARP (Invitrogen, Waltham, MA, USA), cleaved caspase-3, IRE1α, P-p65, p65, P-IKKα/β, IKKα/β, P-IκB (Cell Signaling Technology, Danvers, MA, USA), OGT, p53, p21, PERK, P-ATM, XBP, Bcl2, Bax, lamin, α-tubulin, and β-actin (Santa Cruz, Dallas, TX, USA). After extensive washing with TBS-T, blots were incubated with appropriate horseradish peroxidase (HRP)-conjugated secondary antibodies at room temperature for 30 min. Finally, the protein bands were visualized using an ECL Plus Western Blotting Detection System (GE Healthcare Life Sciences). Signals were quantified by ImageJ software.

### 4.2. In Vivo Xenograft Assay

Five-week-old female BALB/c-Foxn1^nu^ athymic nude mice were purchased from JANVIER Laboratory (France). This study was approved by the Animal Care and Use Committee of the Kyungpook National University, Korea. Approximately 5 × 10^6^ HepG2 cells were harvested and resuspended in a 100 μL mixture of 50% PBS and 50% Matrigel (BD Biosciences) and injected subcutaneously in the flanks of each mouse. One week after xenotransplantation, nude mice with tumors were randomly divided into a low concentration treatment group (0.1 mg/kg of DOX and/or 1 mg/kg of OSMI-1 (Sigma-Aldrich, St Louis, MO, USA) and a high concentration treatment group (1 mg/kg of DOX and/or 5 mg/kg of OSMI-1) and each group was divided into 4 subgroups (*n* = 2 or *n* = 4 per subgroup) as follows: Group 1, mice administered DMSO as vehicle control; Group 2, mice administered DOX; Group 3, mice administered OSMI-1; Group 4, mice coadministered DOX and OSMI-1. Each group was intravenously administered every other day for 4 weeks. The size of subcutaneous tumors was determined using a digital caliper (Mitutoyo, Tokyo, Japan) once every four days. The weight of each mouse was recorded as an indicator of tolerability on these same days. The tumor volumes (V) were measured using the equation V = (L × W^2^)/2 (mm^3^), where tumor length (L) is the largest diameter and tumor width (W) is the perpendicular diameter of L. At the experimental endpoint, mice were sacrificed and the tumor masses were excised and weighed. The tumor masses were fixed in buffered 4% paraformaldehyde and used for immunohistochemical (IHC) staining. The animal experimental procedures were performed in accordance with approved animal protocols and guidelines established by the Animal Care Committee of Kyungpook National University (No. KNU 2019-0002).

### 4.3. Immunohistochemistry

Xenograft tumor tissues were extracted from mice. The tumor tissues were fixed in 4% paraformaldehyde and embedded in paraffin. Sections were deparaffinized and boiled in citrate buffer (pH 6.0) for antigen retrieval. After cooling at room temperature, sections were blocked with 3% BSA containing goat serum for 1 h. The primary antibodies used for histological analysis included cleaved caspase-3, Ki-67, p53, and p65. Sections were then incubated with the appropriate secondary antibodies. The immunohistochemical reactions were visualized using 3, 3-diaminobenzidine (DAB).

### 4.4. Cell Culture and Transfection

The liver cancer cell lines HepG2, Hep3B, and Huh7 were obtained from the Korean Cell Line Bank (KCLB, Seoul, Korea). The murine hepatocyte-derived AML12 cells were provided by Dr. Jae Man Lee (KNUM, Daegu, Korea). HepG2 and Hep3B cell lines were cultured in DMEM, while Huh7 cells were cultured in RPMI-1640 medium supplemented with 10% fetal bovine serum (FBS) and 1% penicillin. The AML12 cell line was cultured in DMEM (F-12) medium containing 10% FBS, 1% penicillin, 1% ITS, and 40 ng/mL dexamethasone. Cells were incubated in a humidified 37 °C and 5% CO_2_ incubator. Control, p53, and OGT siRNA (Santa Cruz, Dallas, TX, USA) were transfected into HepG2 cells using Lipofectamine RNAi MAX (Invitrogen, Waltham, MA, USA). Cells were homogenized in NP-40 lysis buffer and the immunoblot procedure was followed as described above. OGT and p53 primary antibodies (Cell Signaling Technology, Danvers, MA, USA) were used to detect proteins.

### 4.5. Immunoprecipitation (IP)

Total protein extracted from HepG2 cells was incubated with the antibodies against p65 or IKKβ and Dynabeads^TM^ from the Protein G IP kit (Thermo Fisher Scientific, Waltham, MA, USA). Protein was extracted from immune complexes precipitated with protein G by boiling in 2× sample buffer. Protein samples were separated by SDS-PAGE and transferred onto a membrane. Precipitates were analyzed by Western blotting.

### 4.6. Real-Time PCR

Total RNA was extracted using TRIzol reagent (Invitrogen, Waltham, MA, USA). Briefly, lysates were mixed with chloroform then centrifuged at 12,000× *g* at 4 °C for 10 min. The supernatant was mixed with an equal volume of isopropanol. RNA was then pelleted by centrifugation at 12,000× *g* at 4 °C for 10 min. The sample was washed with 70% ethanol twice and RNA was dissolved in DEPC water to a final concentration of 1 μg/μL, which was used for reverse transcription PCR. cDNA was derived using the cDNA synthesis kit (Invitrogen, Waltham, MA, USA). Each qPCR reaction contained 2 μL of stock cDNA, 1 μM of forward and reverse primer mix, and 5 μL of power SYBR Green PCR Master Mix (Thermo Fisher Scientific, Waltham, MA, USA). The PCR primers used are as follows: CHOP, Forward: 5′-AAGGAAAGTGGCACAGCTAGCT -3′, reverse: 5′-CTGGTCAGGCGCTCGATTT -3′; Bcl2, Forward: 5′-TTGTGGCCTTCTTTGAGTTCGGTG -3′, reverse: 5′-GGTGCCGGTTCAGGTACTCAGTCA -3′.

### 4.7. Cell Viability

Cell viability of HepG2 and AML12 cell lines was determined using 3-(4,5-dimethylthiazol-2-yl)-2,5-diphenyl-tetrazolium bromide (MTT). Cells were seeded in 96-well plates at a concentration of 5 × 10^3^ cells/well and the indicated concentration of DOX. The MTT assay was carried out for 15 h following DOX treatments. After the treatment period, cells were briefly washed with PBS. Mixing solution (2 mg of MTT (Sigma-Aldrich, St Louis, MO, USA) in 1 mL PBS) was added to each well and cells were incubated for 4 h at 37 °C. Then, the MTT solution was removed and 100 μL of DMSO was added to each well. The plate was maintained for 30 min at 37 °C and the OD of the wells was determined at 580 nm using a spectrophotometric microplate reader (Molecular Devices, San Jose, CA, USA). Results were expressed as the percentage of cell viability.

### 4.8. Colony Formation Assay

A total of 1000 HepG2 cells were seeded into 6-well plates. The medium was refreshed every 4 days. Cells were harvested after 8–10 days of incubation at 37 °C and washed twice with PBS. Subsequently, colonies were fixed with 100% methanol. Then, the cells were washed with PBS and stained with 0.5% crystal violet. Colonies were counted under an optical microscope. All experiments were performed in triplicate.

### 4.9. Preparation of Whole-Cell and Nuclear Extracts

HepG2 cells were seeded at 1 × 10^5^ cells/well and treated with DOX (Sigma-Aldrich, St Louis, MO, USA) and/or OSMI-1. After treatment, the cells were collected by centrifugation and washed twice with ice-cold PBS. Cytosolic and nuclear proteins were extracted using the NE-PER^®^ Nuclear and Cytoplasmic Extraction Reagents (Thermo Fisher Scientific, Waltham, MA, USA). according to the manufacturer’s instructions.

### 4.10. Statistical Analysis

All data are presented as means ± SD. Two-sample t-tests were performed. For all experiments, *n* = 2–4 animals per group. Significance was established using Mann–Whitney tests, with *p* < 0.05 being considered significant, after correction for multiple comparisons. All statistical analyses were performed using SPSS software.

## 5. Conclusions

The results of this study suggest that the blockade of O-GlcNAcylation signaling directly affects cell survival and resistance to chemotherapy, consequently promoting apoptotic cell death. Moreover, the effectiveness of this synergistic therapy allows the use of smaller doses of a chemotherapeutic agent, further reducing the side effects associated with chemotherapy. Thus, the combination of chemotherapy and OGT inhibition may serve as a potential strategy to improve therapeutic efficacy in various cancers.

## Figures and Tables

**Figure 1 cancers-12-03154-f001:**
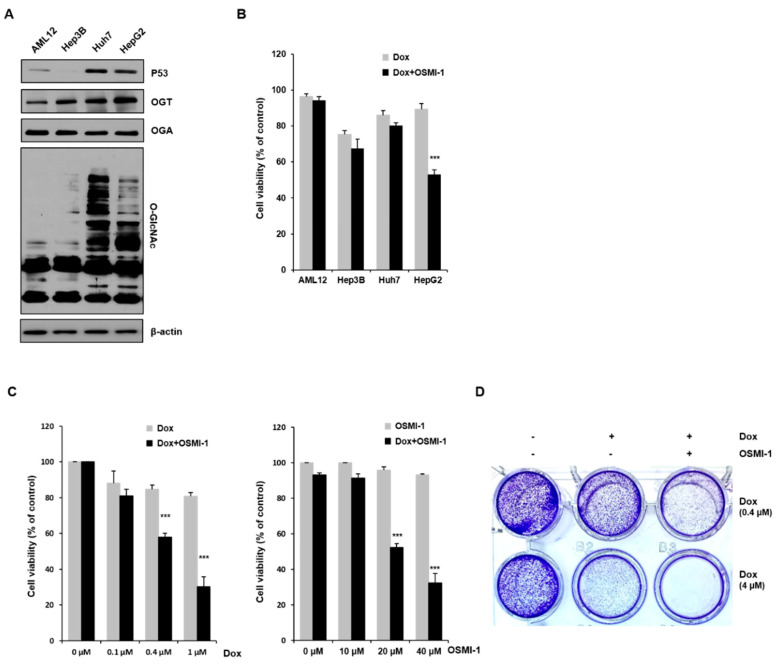
Effects of OSMI-1 and doxorubicin (DOX) on the cell viability of hepatocellular carcinoma (HCC) cells. (**A**) Immortalized human liver cells (AML12) and human hepatocellular carcinoma (HCC) cells (Hep3B, Huh7, and HepG2) were analyzed by Western blot for p53, O-GlcNAc transferase (OGT), O-GlcNAcase (OGA), and O-GlcNAc. β-actin served as a loading control. The graphs on the right show the relative amount of measured protein as fold increase, the uncropped western blot figures in Appendix A. (**B**) AML12, Hep3B, Huh7, and HepG2 cells were treated with 0.4 μM of DOX and 20 μM of OSMI-1 for 15 h, and cell viability was analyzed by MTT assay. *** *p* < 0.005 compared with the DOX-treated group. (**C**) HepG2 cells were treated with various concentrations of DOX (0.1 to 1 μM) in the absence or presence of OSMI-1 (20 μM) for 15 h (left panel). HepG2 cells were treated with various concentrations of OSMI-1 (10 to 40 μM) in the absence or presence of DOX (0.4 μM) for 15 h, and cell viability was analyzed by MTT assay (right panel). Data represent three independent experiments. *** *p* < 0.005 compared with DOX alone or OSMI-1 alone groups. (**D**) A cell colony formation assay was used to determine the proliferation of HepG2 cells in the absence or presence of DOX (0.4 or 4 μM) alone or in combination with OSMI-1 (20 μM). After cultivation for 8–10 days, plates were stained with crystal violet.

**Figure 2 cancers-12-03154-f002:**
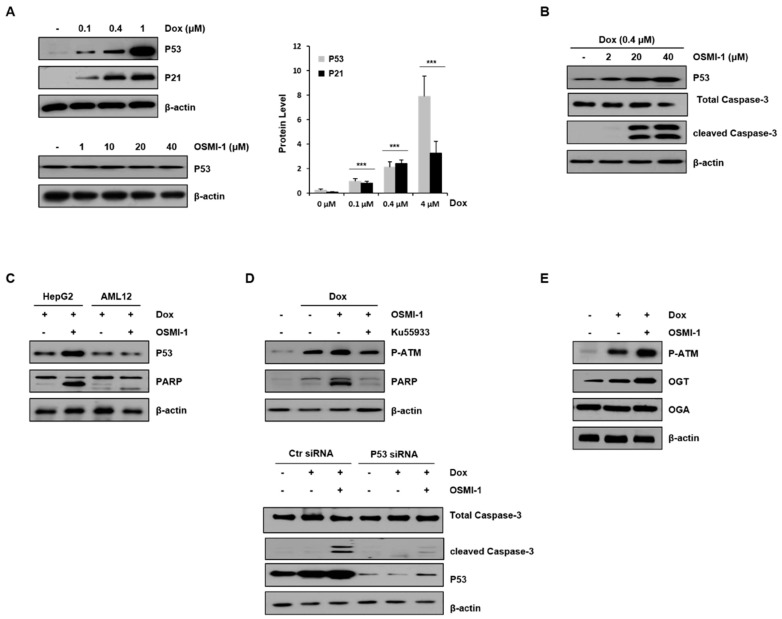
OSMI-1 enhances DOX-induced apoptosis in HepG2 cells via p53 signaling. (**A**) Protein levels of p53 and p21 in HepG2 cells treated with 0, 0.1, 0.4, and 1 μM of DOX for 15 h were analyzed by Western blot (upper panel). The graphs on the right show the relative amounts of protein as fold increase. *** *p* < 0.005 compared with the control group. Similarly, p53 levels in HepG2 cells treated with various concentrations of OSMI-1 (1–40 μM) were analyzed (lower panel). β-actin was used as an internal control. (**B**) DOX (0.4 μM)-induced HepG2 cells were treated with various concentrations of OSMI-1 (2–40 μM) for 15 h. Western blot analysis was performed to determine levels of p53 and cleaved caspase-3. (**C**) HepG2 and AML12 cells were treated with DOX (0.4 μM) alone or in combination with OSMI-1 (20 μM) for 15 h. Levels of p53 and PARP were detected by Western blot. (**D**) The upper panel shows HepG2 cells pretreated with 0.4 μM of DOX and treated with 20 μM of OSMI-1 alone or in combination with 10 μM of ku55933. Cell lysates were analyzed by Western blot using antibodies against phospho-ATM and cleaved PARP. The lower panel shows HepG2 cells after transfection with control or p53 siRNA, which were then treated with DOX (0.4 μM) alone or in combination with OSMI-1 (20 μM) for 15 h. Cell extracts were analyzed for cleaved caspase-3 by Western blot. (**E**) HepG2 cells were treated with DOX (0.4 μM) alone or in combination with OSMI-1 (20 μM) for 15 h. Cell lysates were analyzed by Western blot using antibodies against phospho-ATM, OGT, and OGA. β-actin was used as an internal control. Values are presented as mean ± SD of three independent determinations, the uncropped western blot figures in Appendix A.

**Figure 3 cancers-12-03154-f003:**
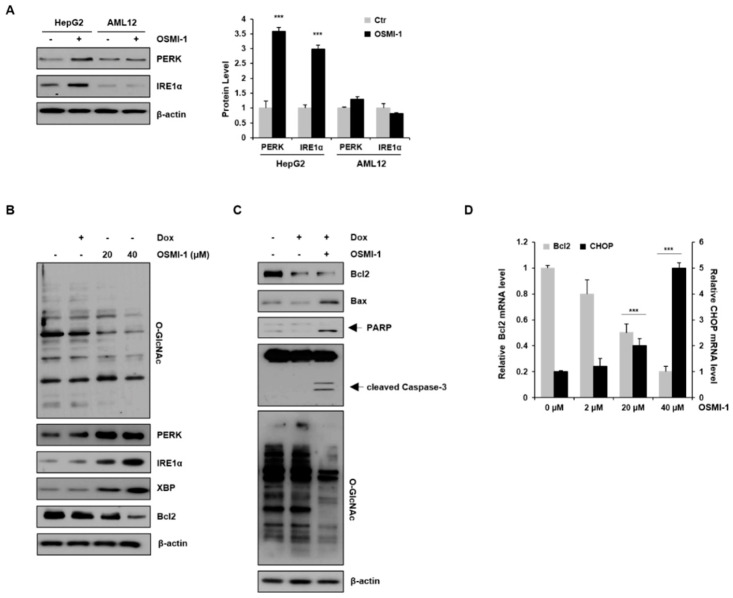
OSMI-1 stimulates the ER stress response. (**A**) HepG2 and AML12 cells were treated with or without 20 μM of OSMI-1 for 15 h, and levels of protein-kinase-R-like endoplasmic reticulum kinase (PERK) and IRE1α were detected via Western blotting. The graphs on the right show the relative amounts of protein as fold increase. *** *p* < 0.005 compared with the control group. (**B**) HepG2 cells were treated with DOX (0.4 μM) or OSMI-1 (20 and 40 μM) for 15 h. Western blot analysis was performed to detect levels of O-GlcNAcylated proteins, PERK, IRE1α, XBP, and Bcl2. (**C**) HepG2 cells were treated with DOX (0.4 μM) alone or in combination with OSMI-1 (20 μM) for 15 h. Cell lysates were analyzed by Western blot using antibodies against Bcl2, Bax, cleaved PARP, caspase-3, and O-GlcNAcylated proteins. β-actin was used as an internal control. (**D**) HepG2 cells were treated with 0.4 μM of DOX alone or in combination with various concentrations of OSMI-1 (2, 20, or 40 μM). Levels of Bcl2 and CHOP mRNA were determined by qRT-PCR and normalized to the control sample. Values represent means ± SD (*n* = 6). *** *p* < 0.005 compared with the control group, the uncropped western blot figures in Appendix A.

**Figure 4 cancers-12-03154-f004:**
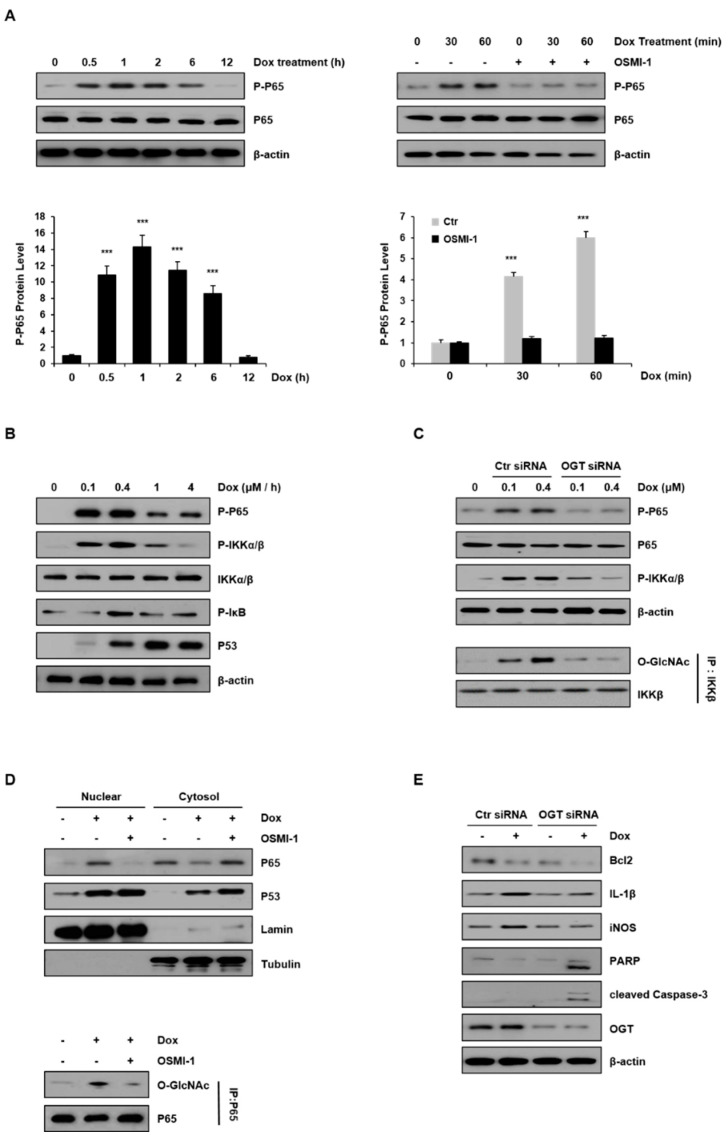
Inflammatory NF-κB signaling was modulated by DOX and OSMI-1. (**A**) Levels of phospho-p65 in HepG2 cells treated with DOX (0.4 μM) were monitored by Western blot for up to 12 h (left). HepG2 cells were treated with DOX (0.4 μM) for up to 60 min in the absence or presence of OSMI-1 (20 μM) and levels of phospho-p65 were analyzed (right). The lower two graphs show the relative amount of these proteins as fold increase. Data represent three independent experiments. *** *p* < 0.005 compared with the control group. (**B**) HepG2 cells treated with DOX (0 to 4 μM) for 1 h were analyzed by Western blot using antibodies against phospho-p65, phospho-IKKα/β, phospho-IκB, and p53. (**C**) After transfection with control or OGT siRNAs, HepG2 cells were treated with DOX (0.1 and 0.4 μM) for 1 h. Levels of phospho-p65 and phospho-IKKα/β in HepG2 cells were analyzed by Western blot. Immunoprecipitation analysis using the IKKβ antibody was performed (lower panel) followed by immunoblotting using the GlcNAc antibody. Immunoprecipitation analysis was performed using whole-cell lysate, and the GlcNAc levels were assessed using the same amount of precipitates. (**D**) OSMI-1 (20 μM) pretreated with or without HepG2 cells were treated with DOX (0.4 μM) for 1 h. The levels of p65 and p53 in the nuclear and the cytoplasmic fractions were analyzed by Western blot. LaminB and tubulin were used as loading controls for the nuclear and cytoplasmic proteins, respectively. Nuclear fractions from the cell lysates were subjected to immunoprecipitation using anti-p65 antibodies, followed by Western blot analysis with anti-O-GlcNAc antibodies (right). (**E**) After transfection with scramble-siRNA or OGT siRNA, HepG2 cells were treated without or with DOX (0.4 μM) for 15 h. Total lysates were subjected to Western blot analysis using the indicated Abs, the uncropped western blot figures in Appendix A.

**Figure 5 cancers-12-03154-f005:**
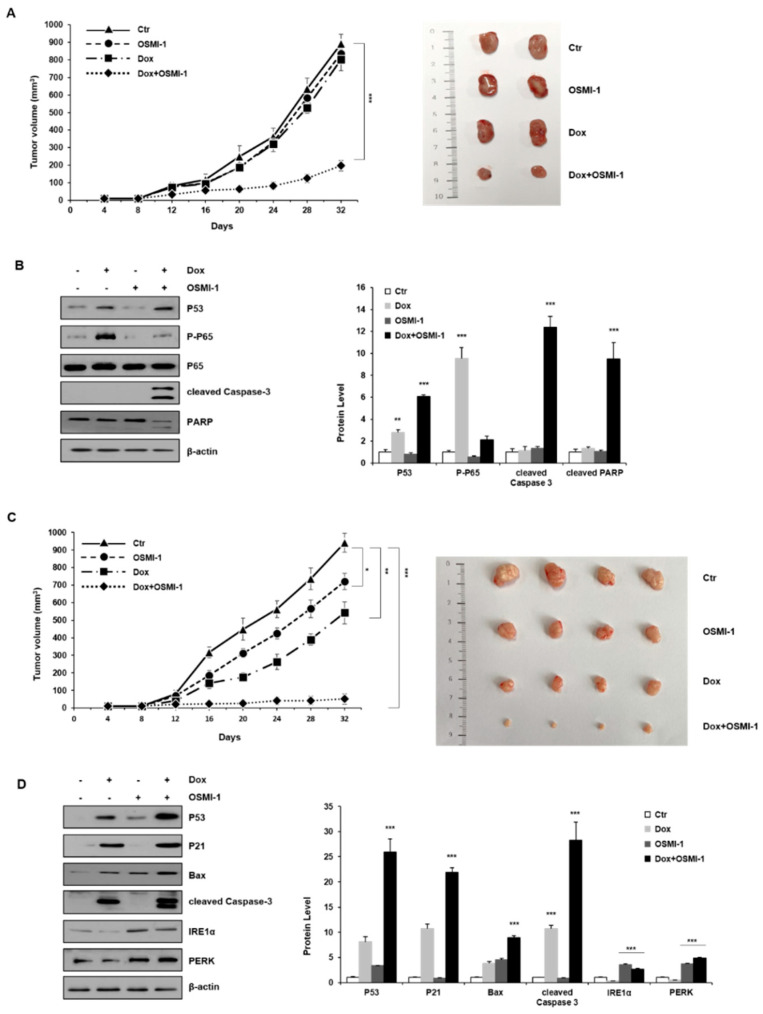
DOX and OSMI combination synergistically inhibited the tumor growth of human HCC xenograft in vivo. (**A**) HepG2 cells were subcutaneously inoculated in nude mice, and tumor growths were monitored for 32 days. Treatments were as follows: vehicle (DMSO, 100 mm^3^), DOX (0.1 mg/kg), and OSMI-1 (1 mg/kg) singly or in combination by IV injection once per 3 days for 3 weeks. A growth profile of each tumor was recorded and expressed as changes in tumor volume (mm^3^) at the indicated time points. Data are shown as mean ± standard deviation of three independent experiments. ** *p* < 0.01 compared with the control group. Representative images of the excised tumors derived from nude mice are shown in the right panel. (**B**) Expression levels of p53, p-p65, p65, cleaved caspase-3, and PARP in the excised tumor tissues derived from nude mice were determined by Western blotting using equivalent amounts of total tissue protein. The expression levels were normalized relative to β-actin. The graphs on the right show the relative amount of measured protein as fold increase. (**C**) As in (A), HepG2 xenograft mice were treated with DOX (1 mg/kg) or OSMI-1 (5 mg/kg) alone or in combination via IV injection, and tumor growth was monitored for 32 days. Volumes of xenograft tumors were measured and plotted at the indicated time points after treatment. Values are presented as means ± SD (*n* = 4). * *p* < 0.05, ** *p* < 0.01, *** *p* < 0.005 compared with the control group. Representative images of the excised tumors derived from nude mice are shown in the right panel. (**D**) Expression levels of p53, p21, Bax, cleaved caspase-3, IRE1, and PERK in the excised tumor tissues derived from nude mice were determined by Western blotting using equivalent amounts of total tissue protein. The expression levels were normalized relative to β-actin. The graphs on the right show the relative amount of measured protein as fold increase, the uncropped western blot figures in Appendix A. (**E**) Sections of the excised tumors treated with DMSO, DOX, OSMI-1, and DOX+OSMI-1 were subjected to immunohistochemical analysis with antibodies against cleaved caspase-3 Ki-67, p53, p65, and PERK (100× and 200× magnification). (**F**) The proposed signaling pathways for therapeutic agents. Arrows and bars represent activation and suppression, respectively. ER stress involves the activation of processes such as apoptosis and inflammation. Modulation of NF-κB, a transcription factor, becomes activated during ER stress sensor IRE1 and PERK to mediate estrogen-induced apoptosis. OSMI-1 blocked NF-κB signaling, resulting in a significant decrease in inflammation by DOX. IKK β can also block cellular p53 stability.

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
