# Peer review of "O-GlcNAc Transferase Inhibitor Synergistically Enhances Doxorubicin-Induced Apoptosis in HepG2 Cells"

_cancers, 2020, doi:10.3390/cancers12113154_

Round 1

Reviewer 1 Report

Le and Kwon in this work investigated the doxorubicin (DOX) and O-GlcNAc transferase (OGT) inhibitor OSMI-1 combination treatment in HepG2 cell lines. Although treatment with a low dose of DOX or of OSMI-1 alone did not induce apoptotic cell death in the cells, the combination synergistically increased apoptotic cell death through activation of both the p53 and mitochondrial Bcl2 pathways in vitro and reduced cell proliferation and tumor growth in vivo as shown using a HepG2 xenograft mouse model. These findings indicate that OSMI-1 could act as a potential chemosensitizer and could be utilized in clinical applications to treat hepatocellular carcinoma.

All experiments have been well conducted and the appropriate controls were used. The data are clear and logically presented, in the study are interesting aspects, also if in its current form the data are too preliminary and do not entirely support the conclusion that should require further validations.

Overall my opinion is positive for the article that should be accepted for publication.

Minor points:

  • MW markers are missing in all western blot. Please provide.
  • Figure 1C: I think it is necessary to explain better the controls.
  • Figure 2: In each WB it is not explain the trend of Caspase-3 but only the cleaved form.

Author Response

Dear Reviewer

Thank you very much for your letter of 19. Oct 2020, regarding manuscript O-GlcNAc transferase inhibitor synergistically enhances doxorubicin-induced apoptosis in HepG2 cells

We have revised the manuscript according to the reviewer's comments and upload the revised file. In addition, as suggested by the editor, the manuscript was adjusted to the structure required by cancers.

The major revisions that we corrected and performed additional experiments are shown below.

Reviewer #1

1) MW markers are missing in all western blot. Please provide.

==> We have added MW markers in all western blot .

2) Figure 1C: I think it is necessary to explain better the controls.

==> We have changed graph in Figure1c control.

3) Figure 2: In each WB it is not explain the trend of Caspase-3 but only the cleaved form.

==> We have added total caspase-3 in Figure2

Reviewer 2 Report

In the current manuscript, entitled-O-GlcNAc transferase inhibitor synergistically enhances doxorubicin-induced apoptosis in HepG2 cells-by Su Jin Le and Oh-Shin Kwon investigated the Combined effect of the doxorubicin (DOX) and O-GlcNAc transferase (OGT) inhibitor OSMI-1 on the cell death in a model of hepatocellular carcinoma using a series of in vitro and in vivo experiments. The authors found that combination treatment reduces cell proliferation and induces apoptosis, indicating the clinical importance of OGT inhibition in treating hepatocellular carcinoma. In this, the authors investigated the role of p53, ER responsive, and NFkB pathways activation and explored the involvement both in vitro and in vivo approaches. Data is presented well and a well-written manuscript. My only question is, how this number of pathways are involved? 

Author Response

Dear Reviewer

Thank you very much for your letter of 19. Oct 2020, regarding manuscript O-GlcNAc transferase inhibitor synergistically enhances doxorubicin-induced apoptosis in HepG2 cells

We have revised the manuscript according to the reviewer's comments and upload the revised file. In addition, as suggested by the editor, the manuscript was adjusted to the structure required by cancers.

The major revisions that we corrected and performed additional experiments are shown below.

My only question is, how this number of pathways are involved? 

==> To explain the mechanisms involved, the following text was added to the discussion section.

ER stress involves the activation of processes such as apoptosis and inflammation. Modulation of NF-κB, a transcription factor, becomes activated during ER stress sensor IRE1 and PERK to mediate estrogen-induced apoptosis. But OSMI-1 blocked NF-κB signaling, resulting in a significant decrease inflammation by DOX. And IKK β can also block cellular p53 stability 

In addition, we have made and marked minor changes, including a sentence in author contribution
